# Research on Three-Dimensional Reconstruction of Ribs Based on Point Cloud Adaptive Smoothing Denoising

**DOI:** 10.3390/s24134076

**Published:** 2024-06-23

**Authors:** Darong Zhu, Diao Wang, Yuanjiao Chen, Zhe Xu, Bishi He

**Affiliations:** 1School of Automation (School of Artificial Intelligence), Hangzhou Dianzi University, Hangzhou 310018, China; zhudr1971@sina.com (D.Z.);; 2Affiliated Hangzhou First People’s Hospital, School of Medicine, Westlake University, Hangzhou 310024, China

**Keywords:** 3D rib reconstruction, point cloud, adaptive smoothing denoising, post-processing

## Abstract

The traditional methods for 3D reconstruction mainly involve using image processing techniques or deep learning segmentation models for rib extraction. After post-processing, voxel-based rib reconstruction is achieved. However, these methods suffer from limited reconstruction accuracy and low computational efficiency. To overcome these limitations, this paper proposes a 3D rib reconstruction method based on point cloud adaptive smoothing and denoising. We converted voxel data from CT images to multi-attribute point cloud data. Then, we applied point cloud adaptive smoothing and denoising methods to eliminate noise and non-rib points in the point cloud. Additionally, efficient 3D reconstruction and post-processing techniques were employed to achieve high-accuracy and comprehensive 3D rib reconstruction results. Experimental calculations demonstrated that compared to voxel-based 3D rib reconstruction methods, the 3D rib models generated by the proposed method achieved a 40% improvement in reconstruction accuracy and were twice as efficient as the former.

## 1. Introduction

3D rib reconstruction technology can help doctors diagnose and treat rib fractures and related diseases, which is of great significance for clinical medicine. Rib fractures are a common finding after chest trauma [1], usually caused by chest trauma, blunt force, and medical conditions such as cancer and obesity [2], in which chest trauma accounts for 10% to 15% of all trauma [3]. Due to inadequate pain control, respiratory complications such as post-traumatic pneumonia often occur due to rib fractures [4]. The structure and morphology of the ribs are a stable reference for various analysis and quantification tasks, such as lung volume estimation and skeletal abnormality quantification [5]. The number and type of rib fractures can serve as the basis for further treatment strategies [6], so accurate detection of rib fractures on CT scans helps with appropriate patient care [7]. 3D rib reconstruction technology can help improve the diagnostic rate of diseases such as rib fractures, and the research and application of this technology is of great significance.

There has been a lot of related research work in the field of 3D rib reconstruction. Traditional 3D rib reconstruction methods mainly rely on image processing, computer vision, and geometric methods. They usually include preprocessing (denoising, enhancement), segmentation (threshold segmentation, region-growing, etc.), and reconstruction (surface-based or voxel-based reconstruction) steps. These methods are greatly affected when dealing with complex backgrounds, noise interference, and incomplete data, resulting in low reconstruction accuracy, and require manual parametric adjustment, with unstable results.

With the development of deep learning technology, 3D rib reconstruction methods based on deep learning have achieved significant performance improvements. These methods usually use neural network models for rib voxel segmentation, which can significantly improve their segmentation accuracy. However, voxel data contain rich position, size, and attribute information, which increases the spatial complexity and computational complexity of reconstruction, greatly affecting the efficiency of reconstruction inference. At the same time, the geometric distortion of an object surface caused by the discrete network structure of voxel segmentation also reduces the accuracy of 3D rib reconstruction.

Therefore, the existing 3D rib reconstruction methods, whether traditional methods or deep learning-based methods, still have limitations in terms of their reconstruction quality, computational efficiency, and inference speed and cannot fully meet the current needs of medical diagnosis and treatment. Designing a method that can obtain stable 3D rib reconstruction results and meet the current medical diagnosis and treatment level in terms of reconstruction quality, reconstruction inference time, and reconstruction accuracy can greatly improve the current medical quality and effectiveness in the rib area.

In this study, we propose a 3D rib reconstruction method based on point cloud adaptive smoothing denoising, which effectively solves the limitations of traditional methods in reconstruction quality and computational efficiency by utilizing the complex structural expression ability and efficiency of point clouds. Our main contributions are summarized as follows. (1) We propose a 3D rib reconstruction method based on point cloud adaptive smoothing denoising, which effectively solves the limitations of traditional methods in reconstruction quality and computational efficiency. (2) We designed a new point cloud denoising algorithm that can effectively eliminate the noise generated in the data acquisition and conversion process, improving the quality and reconstruction accuracy of rib point cloud data. (3) We propose an innovative 3D reconstruction post-processing method that can further optimize 3D rib reconstruction results, improve their reconstruction accuracy and comprehensiveness, and obtain higher-quality 3D rib reconstruction visualization results for medical applications.

The 3D rib reconstruction method proposed in this study not only has technical innovations, but can also bring significant potential impacts to clinical medical practice. More accurate anatomical models can help doctors better diagnose the condition and develop more targeted treatment strategies, thereby improving the success rate of surgery, shortening recovery time, and ultimately benefiting the general public.

## 2. Related Work

Previous research on the 3D reconstruction of bone tissue can be divided into three main categories: traditional image processing methods, deep learning methods, and point cloud reconstruction methods.

Early research on the 3D reconstruction of bone tissue mainly relied on traditional image processing techniques, such as thresholding, region-growing, and edge detection. These methods preprocess original CT images to separate the bone tissue structure and then complete the 3D reconstruction of the bone tissue through 3D surface reconstruction techniques. Chen et al. [8] used sagittal and coronal multiplanar reconstruction (MPR), maximum intensity projection (MIP), and shaded surface display (SSD) to reconstruct the bony structures around the ankle joint and carefully observe the 3D morphology of the ankle joint through MIP. Although this method can achieve good results, it requires parametric adjustment, making the reconstruction process complex. Maken et al. [9] executed the corresponding steps of calibration, contour extraction, 2D image correspondence, and registration in sequence to perform 3D reconstruction from X-ray images and visualize a patient’s 3D anatomical structure. This method can achieve the reconstruction goal, but the steps are cumbersome, require manual intervention, and are time-consuming, making it difficult to obtain stable 3D visualization results. Liu et al. [10] used multiplanar reconstruction CT to calculate the pelvic tilt. This method simplifies the CT reconstruction steps, but is easily affected by noise, limiting the accuracy of the obtained pelvic tilt calculation. Effatparvar et al. [11] proposed a 3D modeling algorithm based on ultrasound and applied this technology to lumbar spine reconstruction. The reconstruction process of this method is relatively simple, but the reconstruction quality is easily affected by the image quality. Gajny et al. [12] proposed a method that relies on a new manual input strategy, using statistical inference, image analysis techniques, and fast manual rigid registration to calculate the parametric model of the spine and obtain a 3D reconstruction model of the spine. Although this method does not require manual intervention, it has many steps and is easy to accumulate errors, affecting the final reconstruction model’s accuracy. In addition, there are some methods based on MRI images [13], which can provide richer soft tissue information and help in the diagnosis and evaluation of spinal diseases. However, this type of method is easily affected by resolution and noise, the obtained reconstruction accuracy is not high, resulting in poor subsequent evaluation effects, and it is also difficult to handle complex morphology and structure.

With the development of deep learning, especially the excellent performance of convolutional neural networks (CNNs) in image processing [14,15], image segmentation [16,17], pattern recognition [18,19], and computer vision [14] tasks, and their applications in the agricultural [20], industrial [21], and medical fields, many studies have begun to explore the use of deep learning methods for the 3D reconstruction of bone tissue. Deep learning methods have strong feature learning capabilities and can automatically learn effective feature representations from data, thereby improving the accuracy of the 3D reconstruction of bone tissue. Kim et al. [22] used the U-Net [23] model to train on 2D images composed of axial and coronal planes from chest CT volumes. The combination of the segmentation masks generated by the model supplemented the spatial information from different planes to reconstruct a 3D rib volume. This method achieves an automated process for the 3D reconstruction of ribs through deep learning, but the voxel-based reconstruction method used has limited reconstruction accuracy and efficiency, with great room for improvement. Aubert et al. [24] proposed a new, fast, and automated 3D spine reconstruction method using a CNN to fit a true statistically shaped model of the spine to the image. This method requires relatively low computational resources, but the obtained spine reconstruction accuracy is limited. Furthermore, Forsberg et al. [25] proposed a method using a deep CNN to segment spinal MRI images, thereby achieving 3D reconstruction of the spine. This method has more stable 3D reconstruction results and improved reconstruction accuracy compared to traditional reconstruction methods, but its reconstruction efficiency is limited due to the low compression and storage efficiency of voxel data when handling large-scale data, the large amount of data to be processed in voxel reconstruction, and the unnecessary calculations and overlapping regions at the voxel edges.

In recent years, 3D reconstruction from point clouds has received widespread attention in the fields of computer vision and computer graphics. These methods involve point cloud preprocessing, segmentation, and 3D reconstruction. The purpose of point cloud preprocessing is to generate a point cloud model or to generate high-quality point clouds for subsequent steps. Typically, this step includes four sub-steps: registration, noise filtering, outlier removal, and downsampling [26]. The purpose of point cloud segmentation is to segment the point cloud and obtain the points that satisfy the mathematical model of the object of interest. With the introduction of PointNet [27] and PointConv [28] into point cloud segmentation, the performance of point cloud segmentation has been further improved. 3D reconstruction from point clouds is different from mesh reconstruction [29], and its purpose is to generate a geometric model of the object of interest based on the point cloud segments obtained in the previous steps, effectively generating the object shape in the form of a dense point cloud. Current research has applied 3D point cloud reconstruction to medical imaging. Wu et al. [30] analyzed point clouds widely used in the medical field and improved the neighbor aggregation module in point cloud analysis networks to enhance their ability to extract complex biological structures. Wang et al. [31] proposed an effective method for facial reconstruction using depth sensors. For each test subject, multiple views of the 3D point cloud are acquired, and these point cloud sets are registered using the Iterative Closest Point (ICP) algorithm to reconstruct the 3D model. Although this method can achieve facial reconstruction, it lacks an effective point cloud denoising method, resulting in limited reconstruction accuracy and quality. Dixit et al. [32] used machine learning methods to extract features such as color, size, and depth from 2D X-ray images of the femur and used this information to create a mesh point cloud, then converted the image into an STL representation and created a 3D femur model using a CNN. This method obtained point cloud data with rich information and avoided information loss during the conversion process, but lacked effective post-processing methods, resulting in limited reconstruction accuracy. Stiles et al. [33] proposed a new deep learning-based framework that can automatically reconstruct the 3D left atrial surface directly from the point clouds obtained from the mapping systems commonly used in clinical practice. This CNN-based method has a lightweight design and can complete the reconstruction for each patient in about 10 s. Independent testing on two cross-modal clinical datasets showed that the method achieved a Dice coefficient of 93% and a Hausdorff distance of less than one pixel. This method can directly process 3D data with high spatial resolution and efficiency, but the process of acquiring point cloud data introduces noise and loss of detailed information, which reduces the accuracy of left atrial reconstruction. Banerjee et al. [34] proposed a fully automatic surface reconstruction pipeline that can reconstruct 3D cardiac anatomical meshes of multiple classes from raw cine MRI acquisitions. A key component is a multi-class point cloud completion network (PCCN) that can correct the sparsity and misalignment problems in 3D reconstruction tasks within a unified model. Compared to the benchmark 3D U-Net [35], this method reduced the reconstruction error by 32% and 24%, respectively. This method uses a PCCN to improve the reconstruction accuracy, but it increased the inference time, which was not conducive to clinical application. In addition, Yang et al. [36] first applied point clouds to the field of rib segmentation, converting voxels to point clouds and using the PointNet++ model to segment the rib point clouds from CT chest images, achieving accurate and stable rib point cloud segmentation. This method provides a knowledge base for the 3D reconstruction of rib point clouds and offers new ideas for this field, but is limited by the single point cloud denoising method and the limited feature extraction capability of the point cloud segmentation model, resulting in a bottleneck in rib segmentation accuracy, which is not conducive to obtaining high-quality 3D rib reconstruction results.

In summary, traditional image processing methods have the drawbacks of cumbersome steps and unstable visualization results in the field of skeletal 3D reconstruction, while deep learning methods have the drawbacks of a long computation time and limited accuracy. The 3D reconstruction results obtained by both methods are difficult to meet the requirements of the medical field. 3D reconstruction from point clouds avoids these cumbersome steps by leveraging the advantage of no manual segmentation, and point cloud data can provide detailed surface information of objects and have high efficiency, which helps to quickly generate high-precision 3D models. Therefore, in this paper, we used high-efficiency and expressive point cloud data instead of the voxel data used in previous studies and achieved the 3D reconstruction of ribs through an accurate and automated reconstruction process.

## 3. Method

To overcome the limitations of existing methods for the 3D reconstruction of ribs, including their low accuracy and inefficiency, this paper proposes a novel method for 3D rib reconstruction based on the adaptive smoothing and denoising of point clouds. As shown in Figure 1, the overall framework of this method consists of three parts. The first part is the multi-attribute point cloud data acquisition module, which obtains point cloud data that capture the shape and structure of the object, rather than just pixel intensity information. Specifically, the voxel grid in the CT image is filtered based on Hounsfield unit (HU) values to obtain rib voxel grid data, which are then converted to point cloud data represented by spatial coordinates. The normal vectors of the rib surface are also obtained to enrich local information on the point cloud data. The second part is the point cloud adaptive smoothing and denoising module, which uses noise removal algorithms to eliminate noise generated during data conversion. The point cloud data are also segmented using a semantic segmentation model to obtain accurate and stable rib point cloud data. The third part is the 3D reconstruction and post-processing module, which uses 3D reconstruction techniques to obtain a 3D model of the rib and performs post-processing operations such as outlier removal, point cloud data smoothing, and label smoothing to further improve the quality and accuracy of the reconstruction results.

### 3.1. Acquisition of Multi-Attribute Point Cloud Data

Currently, there are no publicly available point cloud datasets for human tissue reconstruction. In this study, we analyzed the distribution of Hounsfield unit (HU) values in human rib tissue and used a HU threshold range of [200, 1000] to select voxel data containing ribs, which were then used to generate a voxel grid containing rib voxels. We used a topology-based point cloud generation algorithm to process rib data with complex geometric shapes and obtain point cloud data with high accuracy and completeness, as shown in Figure 2. The generated point cloud data include spatial coordinate and normal vector attribute information in the format (x, y, z, nx, ny, nz) compared to point cloud data with only spatial coordinate attribute information in the format (x, y, z). This not only reduces the loss of information from voxel-to-point cloud conversion to enhance the local geometric representation of point cloud data, but also helps to display clearer rib contours during 3D reconstruction. After generating the point cloud data, we matched the mask data with the point cloud data point-by-point and saved them as point cloud labels, providing useful supervised information for subsequent point cloud segmentation model training. Additionally, we converted the rib mask into point cloud data as standard rib point cloud data for subsequent experimental evaluation.

Through the above process, we successfully converted human voxel data into multi-attribute point cloud data and performed corresponding preprocessing work, laying the foundation for subsequent point cloud adaptive smoothing and denoising, 3D reconstruction, and post-processing operations. In the following sections, we will describe these processing methods in detail to achieve high-precision and comprehensive 3D reconstruction results for ribs.

### 3.2. Point Cloud Adaptive Smoothing Denoising

In the process of the 3D reconstruction of ribs, the point cloud data obtained from CT scans often come with noise and errors. In order to improve the accuracy, visualization, and analysis efficiency of subsequent rib modeling, this study adopted an outlier removal method to process rib point cloud data, specifically using a combination of outlier removal and PointTransformer model segmentation to obtain high-quality rib point cloud data.

By using Formula (1), we obtained n point cloud data that needed to be denoised. Using the KNN method, we adaptively selected neighboring points for each point and used Formula (2) to obtain K neighboring points for each point.
(1)$$p=\left\{p_{1},p_{2,}p_{3},\ldots ,p_{n}\right\}$$
(2)$$p_{i}=\left\{p_{i1},p_{i2},p_{i3}\ldots ,p_{ik}\right\}$$

Using Formula (3), we calculated the number of neighboring points to obtain the point density $c_{i}$ for each point. Then, using Formula (4), we calculated the average distance $d_{i}$ between each point and its neighboring points.
(3)$$c_{i}=k,\;\;\;\;\;\;\;\;\;\;\;\;\;\;\;\;\;\;\;\;\;\;\;\;\;\;\;\;\;\;\;\;\;\;\;\;\;\;\;\;\;\;\;\;\;\;\;\;\;\;\;i=1,2,\ldots ,n$$
(4)$$d_{i}=\frac{1}{k}\sum_{j=1}^{k}\left||p_{i}-p_{ij}|\right|,\;\;\;\;\;\;\;\;\;\;\;\;\;\;\;\;\;\;\;i=1,2,\ldots ,n$$

By setting thresholds for the point density and average distance, we used Formula (5) to determine whether each point satisfied the following conditions: the point density $c_{i}$ was greater than the point density threshold $c_{th}$, and the average distance $d_{i}$ was smaller than the average distance threshold $d_{th}$. For points that did not meet these conditions, we considered them as noise points and removed them.
(5)$$c_{i}>c_{th}\;\;\;\;and\;\;\;\;\;d_{i}<d_{th}$$

After removing the noise points, we use the Gaussian kernel smoothing method to smooth the point cloud data and used Formula (6) to obtain the optimized point cloud data.
(6)$${\;\;\;\;\;\;\;\;\;\;p}_{i}^{\prime \prime }=\frac{\sum p_{j}^{\prime }\in N(p_{i}^{\prime })e^{\frac{{-||p_{i}^{\prime }-p_{j}^{\prime }||}^{2}}{2\sigma^{2}}}p_{j}^{\prime }}{\sum p_{j}^{\prime }\in N(p_{i}^{\prime })e^{\frac{{-||p_{i}^{\prime }-p_{j}^{\prime }||}^{2}}{2\sigma^{2}}}},\;\;\;\;\;\;\;i=1,2,\ldots ,n$$
where $p_{i}^{\prime \prime }$ is the coordinate of the point cloud after Gaussian kernel smoothing, N($p_{i}^{\prime }$) is the set of neighboring points of the point $p_{i}^{\prime }$, $p_{j}^{\prime }$ is one of the neighboring points, σ is the width of the Gaussian kernel, set to 2 to obtain a good smoothing effect while preserving details, and $\left||p_{i}^{\prime }-p_{j}^{\prime }|\right|$ is the distance between points $p_{i}^{\prime }$ and $p_{j}^{\prime }$. Additionally, in this study, we used the PointTransformer point cloud semantic segmentation model [37] to extract the rib region from the point cloud data after the above process, removing residual noise points and non-rib point clouds to obtain accurate and high-quality rib point cloud data.

As shown in Figure 3, during the inference stage, a single test sample CT image was converted to point cloud data and processed with point cloud adaptive smoothing and denoising. Specifically, after removing noise points to obtain the point cloud data to be segmented, we sampled the point set in batches of 50 K points and shuffled the point sets to comply with the unordered principle of point clouds. The shuffled point sets were then fed into the PointTransformer loaded with the best weights to obtain the predicted rib point cloud. The predicted results of each point set were visualized using partial point cloud labels. Finally, all the partial point cloud predictions were merged to obtain the complete rib point cloud prediction. Multiple predictions were made using majority voting to improve prediction accuracy. After obtaining the complete point cloud prediction, 3D reconstruction and post-processing operations were performed to obtain the 3D reconstruction results of the ribs.

### 3.3. 3D Reconstruction and Post-Processing

During the reconstruction process, we used a point-based surface reconstruction algorithm, which can generate accurate and smooth 3D models of ribs. After reconstruction, we post-processed the generated 3D models to eliminate any errors introduced during the reconstruction process and obtain high-precision and comprehensive 3D reconstruction results for ribs.

Firstly, for each point, we used Formulas (7) and (8) to calculate the average distance $u_{i}$ and standard deviation $\sigma_{i}$ of its k-nearest neighbors and used Formula (9) to determine whether each point was an outlier by comparing its distance with the sum of the average and a certain multiple of the standard deviation. All data judged to be outliers were removed from the predicted point cloud.
(7)$$u_{i}=\frac{1}{k}\sum p_{j}\in N(p_{i})||p_{i}-p_{j}||$$
(8)$$\sigma_{i}=\sqrt{\frac{1}{k}\sum p_{j}\in N(p_{i}){\left(||p_{i}-p_{j}||-\mu_{i}\right)}^{2}}$$
(9)$$||p_{i}-\mu_{i}||>t*\sigma_{i}$$

Secondly, we converted the predicted point cloud data and corresponding labels to 3D matrices with the same size as the original CT image, determined the number of connected regions in the matrix, and removed smaller connected regions to eliminate non-rib structures. We used morphological closing operations (dilation followed by erosion) to smooth the segmented rib point cloud data. Finally, we set a filter window size, counted the number of occurrences of predicted point cloud labels within the window, and selected the label value with the highest occurrence as the mode of the window to smooth misclassified points in the rib point cloud prediction labels.

## 4. Experiment

To validate the effectiveness of the proposed 3D rib reconstruction method based on point cloud adaptive smoothing and denoising, we designed a series of experiments to evaluate its performance. The experiments mainly consisted of the following parts.

### 4.1. Dataset

In this study, we used the publicly available CT rib datasets RibFrac [38] and RibSeg [36], which contain CT images of patients of different ages, genders, and disease states. The mask files in RibSeg were labeled with a uniform voxel value of 1 for ribs. However, to distinguish between different ribs during 3D reconstruction, we needed to label ribs with different voxel values. Therefore, we uniformly labeled each rib in the mask file. Firstly, we analyzed the connected domains in the CT images to obtain their centroid coordinates and identified whether they were left or right ribs based on the X component of the centroid coordinates. Secondly, we determined the order of the ribs based on the Y and Z components of the centroid coordinates and labeled each rib with a unique voxel value (1–12 and 13–24) in the order of the ribs. Finally, we visually inspected the labeled voxels for each rib and corrected any labeling errors to ensure accurate annotation.

As shown in Figure 4, we obtained a mask file with 24 ribs labeled with different voxel values based on the above labeling method. We preprocessed these images, including HU value filtering, to generate point cloud data suitable for this experiment. As shown in Table 1, the dataset was divided into training, validation, and testing sets, with a reasonable distribution of 8:1:1 for the three sets to evaluate the performance of the model.

### 4.2. Experimental Setup

In terms of the training settings for rib segmentation, considering the dependence on hardware resources of the input size and batch size, the number of points downsampled per iteration during training was set to 10 K. We used the AdamW optimizer for training, which is a high-performance optimization algorithm that combines the advantages of momentum update and regularization, and performs excellently when training deep learning models. The total training cycle was 300, and the batch size was 8. The learning rate update method adopted a warm-up cosine annealing approach, where the warm-up rounds were 15, the maximum learning rate was 0.1, and the minimum learning rate was 0.00001. The warm-up cosine annealing learning rate strategy can achieve rapid convergence in the early stage of training and gradually decrease the learning rate in the later stage, which helps improve the stability and convergence of model training. Setting the minimum learning rate to 0.00001 ensures that the model still has sufficient learning ability in the later stage of training and does not get stuck in a local optimum. The maximum learning rate of 0.1 ensures that the model has a relatively fast convergence speed in the early stage of training. Through this learning rate scheduling strategy, we hoped to achieve rapid convergence and good final performance during the training process.

### 4.3. Evaluation Metrics

To evaluate the method proposed in this paper, we compared it with the traditional voxel-based 3D rib reconstruction method. An inference test was performed on a machine with a Linux operating system and four GeForce 3090Ti GPUs, using PyTorch 1.11.0 and Python 3.9. In the experiment, we focused on the following evaluation metrics: reconstruction quality, reconstruction accuracy, and reconstruction time. To evaluate the quality of the 3D rib reconstruction, we calculated the Completeness Ratio (CR) between the reconstructed ribs and the ground truth ribs to assess the reconstruction quality of individual test samples. The CR is typically calculated by comparing the overlap between the point cloud or 3D model of the reconstruction result and the standard point cloud or ground truth model. It measures the percentage of the true object surface or point cloud that is contained in the reconstruction result. The closer the CR is to 100%, the more of the true object’s surface or point cloud is included in the reconstruction result, indicating higher reconstruction quality. To measure the difference between the reconstructed ribs and the ground truth ribs, we calculated the Root Mean Square Error (RMSE) to evaluate the reconstruction accuracy of individual test samples:(10)$$RMSE=sqrt(\sum \frac{{\left(P_{i}-Q_{i}\right)}^{2}}{N})$$
where $P_{i}$ and $Q_{i}$ represent the coordinates of the i-th point on the reconstructed ribs and true ribs, respectively, and N is the total number of paired points. A smaller RMSE value indicates higher reconstruction accuracy.

To measure the time required for rib reconstruction, we calculated the time required for the complete execution of the reconstruction method to evaluate the reconstruction time of a single test sample.

### 4.4. Experimental Results

To verify the effectiveness of the point cloud adaptive smoothing and denoising method described in this paper, and to compare the effects of different parameters on point cloud adaptive smoothing and denoising, we used the pre-denoised point cloud data as the experimental object. By setting different point density thresholds and average distance thresholds, and setting a random seed during the use of the semantic segmentation model to ensure the invariance in its segmentation prediction, we finally obtained a denoised rib point cloud. To evaluate the denoising effect, we calculated the Mean Root Mean Square Error (Mean RMSE) between the denoised rib point cloud of all test samples and the standard rib point cloud. To obtain the optimal point density threshold and average distance threshold, this experiment tested the Mean RMSE under different combinations of point density thresholds (0.05, 0.10, 0.15, 0.20, 0.25, 0.30) and average distance thresholds (1, 2, 3, 4, 5, 6, 7, 8). The experimental data were used to plot the 3D surface graph shown in Figure 5, which helped us intuitively understand the trend of the denoising performance of the point cloud adaptive smoothing and denoising method under different parametric configurations. By observing this 3D surface graph, the trend of the influence on the Mean RMSE was an upward-opening parabolic trend, and the area with a relatively low Mean RMSE was around Tp = 0.15, which was helpful in finding the combination of Tp and Td that minimized the Mean RMSE, thereby optimizing the performance of the point cloud adaptive smoothing and denoising method and obtaining the best denoising effect. To more objectively evaluate the significance and reliability of the experimental results, as shown in Table 2, we compared the Mean Completeness Ratio (Mean CR) and Mean RMSE obtained under three different average distance thresholds (4, 5, 6) with the point density threshold set to 0.25, 0.15, and 0.05, respectively. The results show that when the point density threshold was 0.15, the overall Mean CR was higher than 0.25 and 0.05, and the Mean RMSE was generally lower than 0.25 and 0.05. Additionally, when the point density threshold was fixed, the best performance was obtained when the average distance threshold was set to 5.

As shown in Figure 6a, one of the test samples was selected for visualization to observe the human body point cloud model before denoising. Then, the outlier removal step in our point cloud adaptive smoothing and denoising algorithm was used to process the noisy points, with the three sets of point density thresholds and the corresponding best average distance thresholds as described above. As shown in Figure 6b, when the point density threshold and average distance threshold were set to 0.25 and 5, respectively, bone tissue fracture occurred, as indicated by the red circle, indicating an over-denoising state. As shown in Figure 6c, when the point density threshold and average distance threshold were set to 0.05 and 5, respectively, noise point attachment occurred, as indicated by the red circle, indicating an under-denoising state. As shown in Figure 6d, when the point density threshold and average distance threshold were set to 0.15 and 5, respectively, the bone structure was clear and complete, and the best denoising effect was obtained.

To verify the effectiveness of the research methods presented in this paper, we conducted a performance evaluation by comparing three different methods for 3D rib reconstruction. The first method is based on the voxel-based U-Net segmentation model from [22] to segment ribs and then reconstruct the 3D rib volume from the segmentation results. The second method is based on the point cloud segmentation model PointNet from [36] to segment ribs, followed by post-processing to achieve 3D rib reconstruction. The third 3D rib reconstruction method is based on the PointTransformer model used in this study for rib segmentation, followed by post-processing to achieve 3D rib reconstruction, but without using the outlier removal and post-processing methods in the point cloud adaptive smoothing and denoising algorithm proposed in this paper. We applied the above three 3D rib reconstruction methods and our proposed method to generate 3D rib models for all testing set CT images, and then calculated the corresponding cr, rmse, and time metrics, as well as the Mean CR, Mean RMSE, and Mean TIME. As shown in Table 3, the voxel-based 3D rib reconstruction method achieved the lowest Mean CR, the highest Mean RMSE, and the longest Mean TIME, fully demonstrating the defects of using voxel-based segmentation for 3D rib reconstruction. The three point cloud-based 3D rib reconstruction methods, on the other hand, were able to achieve a higher Mean CR, lower Mean RMSE, and faster Mean TIME, demonstrating the efficiency of point cloud-based methods. Among the point cloud-based methods, the PointNet model had limited feature extraction capabilities, resulting in a lower Mean CR and higher Mean RMSE, making it difficult to achieve high-precision and high-quality 3D rib reconstruction. In comparison, the transformer-based model introduced in this study, with its self-attention mechanism, can better capture global information and relationships between different points in the point cloud, enabling more accurate rib segmentation and better capture of the surrounding thoracic structure’s contextual information. This led to a 1 percentage point improvement in the Mean CR and a 75 percentage point reduction in the Mean RMSE, resulting in improved reconstruction quality and accuracy. When using the method proposed in this chapter, which introduced point cloud outlier removal and post-processing on top of the PointTransformer model, point cloud denoising first improved the feature accuracy of each point, allowing the PointTransformer to better capture useful structural information. Additionally, the removal of outliers and noise in the input data helped reduce the propagation of errors, simplified the learning task, and accelerated subsequent inference. As a result, the Mean CR was improved by 3 percentage points, and the Mean RMSE was reduced by 69 percentage points, significantly outperforming the other three methods, while the Mean TIME was not noticeably increased.

The distribution statistics of the cr, rmse, and time values obtained for the testing set CT images using the four methods, as shown in Figure 7, Figure 8 and Figure 9, further demonstrate the effectiveness of the proposed method. Compared to the other methods, the CR values obtained using the proposed method were consistently higher, the RMSE values were consistently lower, and the reconstruction time was on par with the other point cloud-based methods. Additionally, the variances in these three metrics were smaller for the proposed method, indicating more stable and consistent results, reducing the uncertainty in clinical applications. These observations provide strong evidence of the effectiveness of the proposed method and further verify its excellent performance in the task of 3D rib reconstruction. This offers valuable support for research and practical applications in this field and provides a useful reference for improving 3D rib reconstruction techniques.

We conducted a visual analysis of the results of four different three-dimensional rib reconstruction methods, as shown in Figure 10. From these visualization results, we can clearly observe a gradual improvement in the reconstruction quality and a reduction in reconstruction errors across the four methods.

Specifically, the first method produced relatively low-quality reconstruction results, with numerous misclassified points on each rib and an incomplete rib structure, resulting in an overall unsatisfactory visual effect. As the methods were progressively improved, the reconstruction quality of the last three methods became increasingly higher. The results of the second method were noticeably better than the first, with fewer misclassified points and a more complete rib structure. The results of the third method further improved, with even fewer misclassified points and clearer rib structures. By the time we reached the fourth method, the overall model visualization reached the best state, with each rib accurately reconstructed, no obvious misclassified points, and a very clear and complete three-dimensional reconstruction model. This series of visualization results fully demonstrates the advantages of the four-rib three-dimensional reconstruction methods proposed in this chapter in terms of practicality and effectiveness. These methods not only can accurately and efficiently achieve the three-dimensional reconstruction of ribs, but they will also play an important role in future clinical applications, providing valuable three-dimensional reference information for medical diagnosis and treatment.

### 4.5. Ablation Experiment

To verify the effectiveness of the post-processing method described in this paper, we conducted ablation experiments on the post-processing method. As shown in Table 4, in order to compare the effects of each stage of the post-processing method on the three-dimensional reconstruction and post-processing of the point cloud, we used the point cloud data before post-processing as the test object and obtained the post-processing point clouds at different stages. We calculated the mean square error between the rib point clouds obtained at each stage and the standard rib point clouds for all testing set samples and then evaluated the post-processing effect by taking the average mean square error. Without any post-processing, the quality of the obtained rib point clouds was relatively low, with many outliers, irregular regions, and misclassified points, resulting in a lower Completeness Ratio (CR) and higher Root Mean Square Error (RMSE). After outlier removal, a large number of outliers were eliminated, improving the quality of the rib point clouds, with the CR increasing by 1 percentage point and the RMSE decreasing by 12 percentage points. After point cloud data smoothing, the holes formed in the predicted rib regions were filled, resulting in a smoother rib region, with the CR increasing by 1 percentage point and the RMSE decreasing by 10 percentage points. Finally, after point cloud label smoothing, the numerous prediction errors in the smoothed rib point cloud were greatly improved, with the CR increasing by 1 percentage point and the RMSE decreasing by 11 percentage points. These results confirm the effectiveness of the proposed post-processing method in improving the quality, accuracy, and completeness of the reconstructed rib point clouds.

### 4.6. Post-Processing Visualization Analysis

As shown in Figure 11, we visualized the post-processing effect on a test sample that had undergone point cloud adaptive smoothing and denoising. This sample had some misclassified points, which were either non-rib point clouds misclassified as rib point clouds, such as some colored point clouds above the left and right ribs, or a single rib point cloud predicted as another rib point cloud, such as some other colored point clouds on each rib. After the post-processing algorithm, the isolated misclassified points above the ribs were removed, and the predicted labels of the point cloud data on each rib were smoothed out, greatly enhancing the visualization effect of the ribs.

### 4.7. Limitations and Future Research Directions

The current research achieved certain results in three-dimensional rib reconstruction, but for some specific application scenarios, such as precise surgical planning or prediction, further improvement in the precision and realism of this reconstruction may still be needed. To this end, future research can focus on the following aspects:

(1) Explore more accurate reconstruction algorithms. Techniques such as machine learning or deep learning can be studied, leveraging the characteristics of point cloud data to develop more accurate three-dimensional rib reconstruction methods. Additionally, the use of additional medical imaging data, such as MRI or CT, can be considered to further improve the precision of the reconstruction results.

(2) Expand to other types of medical imaging. The current research was primarily based on point cloud data, and in the future, the proposed adaptive smoothing and denoising algorithm could be extended to other types of medical imaging data, such as CT and MRI, to explore its three-dimensional reconstruction performance in different imaging modalities. This will help the method to play a role in a wider range of medical applications.

(3) Improve computational efficiency. The current reconstruction process may have efficiency issues when dealing with large-scale point cloud data. To meet the demands of real-time or near-real-time applications, such as robotic control, future research can explore algorithmic optimization or parallel computing techniques to increase the speed of point cloud adaptive smoothing, denoising, and three-dimensional reconstruction.

Through in-depth research in the above directions, we believe we can further improve the precision and efficiency of three-dimensional rib reconstruction, providing support for more medical application scenarios and driving the further development of this field.

## 5. Conclusions

This paper proposes a three-dimensional rib reconstruction method based on point cloud adaptive smoothing and denoising, addressing the issues of low quality, insufficient accuracy, and low efficiency in traditional voxel-based rib reconstruction methods. By leveraging the advantages of point clouds over voxels in terms of data precision and processing speed, and by utilizing appropriate point cloud denoising algorithms to significantly remove non-rib point clouds, the proposed method then employs effective post-processing for three-dimensional reconstruction to obtain complete and accurate rib point clouds, thereby achieving superior reconstruction quality, precision, and computational efficiency. The experimental results confirmed the effectiveness and practicality of this method in the field of three-dimensional rib reconstruction, providing a new solution for the related field. In future research, we will continue to optimize the model’s structure and parameters to further improve its reconstruction quality, precision, and computational efficiency, while also exploring the possibility of applying this method to a wider range of medical imaging tasks.

## Figures and Tables

**Figure 1 sensors-24-04076-f001:**
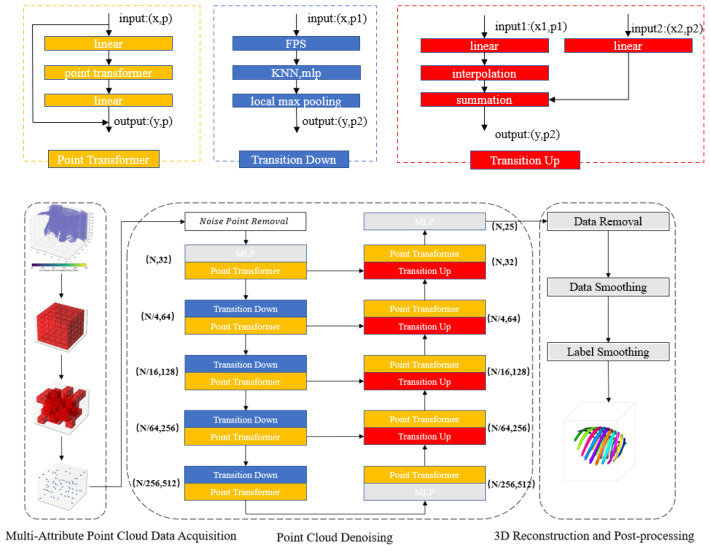
Framework of 3D Rib Cage Reconstruction Method Based on Point Cloud Adaptive Smoothing Denoising.

**Figure 2 sensors-24-04076-f002:**
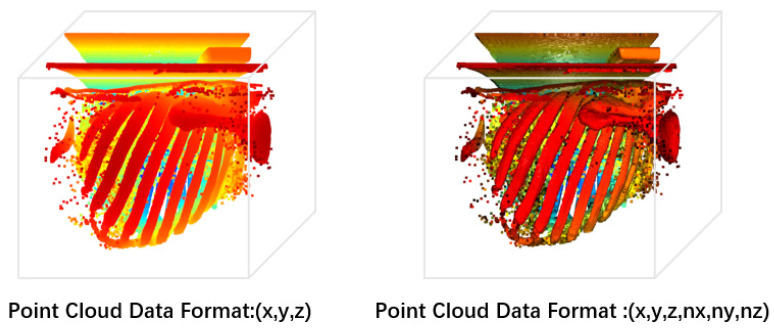
Comparison of Point Cloud Data Attribute Effects.

**Figure 3 sensors-24-04076-f003:**
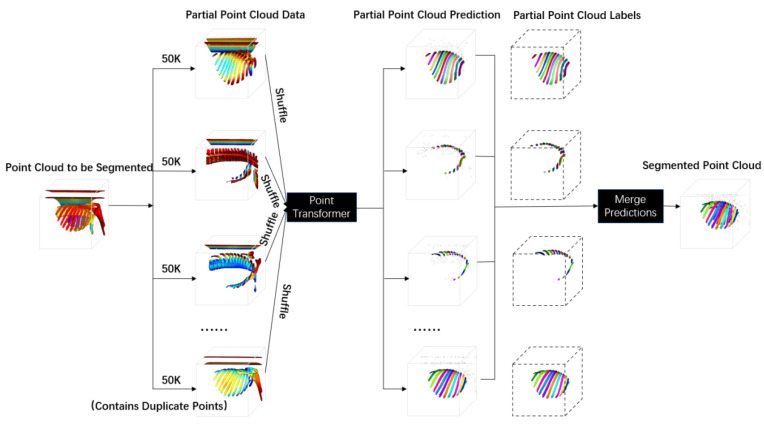
Schematic Diagram of Point Cloud Inference Stage.

**Figure 4 sensors-24-04076-f004:**
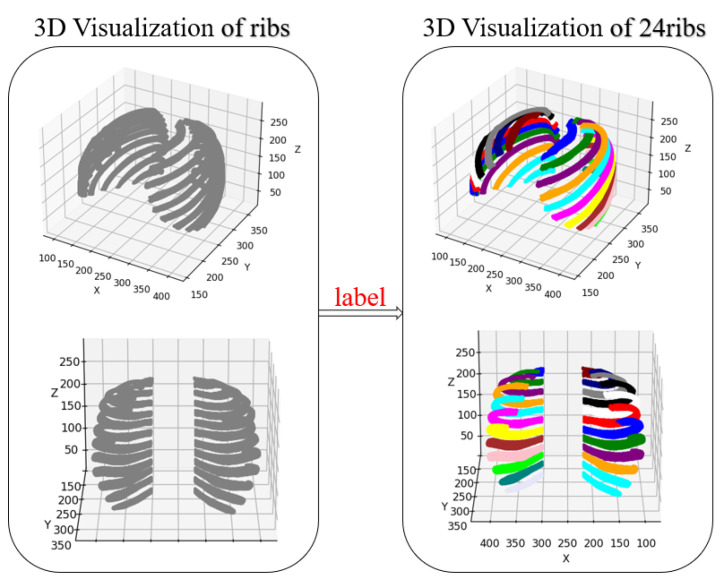
Schematic Diagram of Independent Labeling of Ribs.

**Figure 5 sensors-24-04076-f005:**
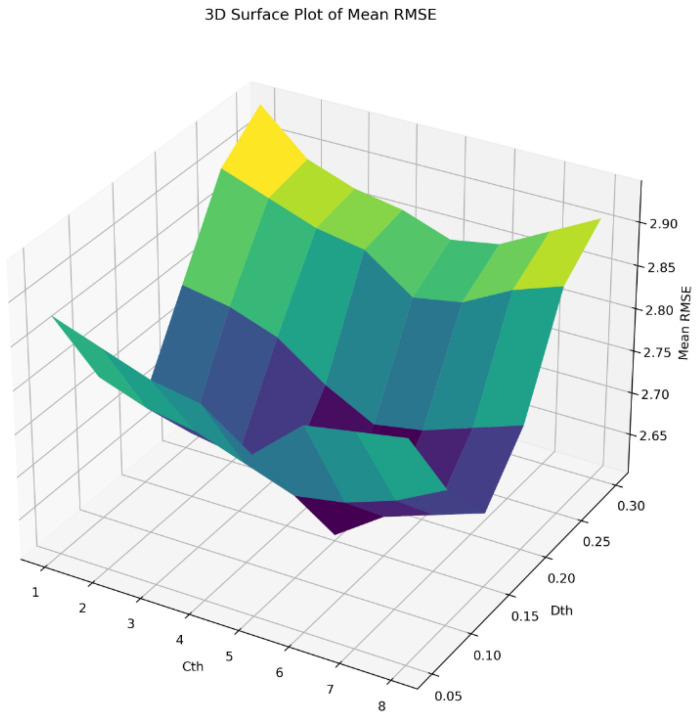
3D Surface Graph of the Point Cloud Adaptive Smoothing Denoising Algorithm.

**Figure 6 sensors-24-04076-f006:**
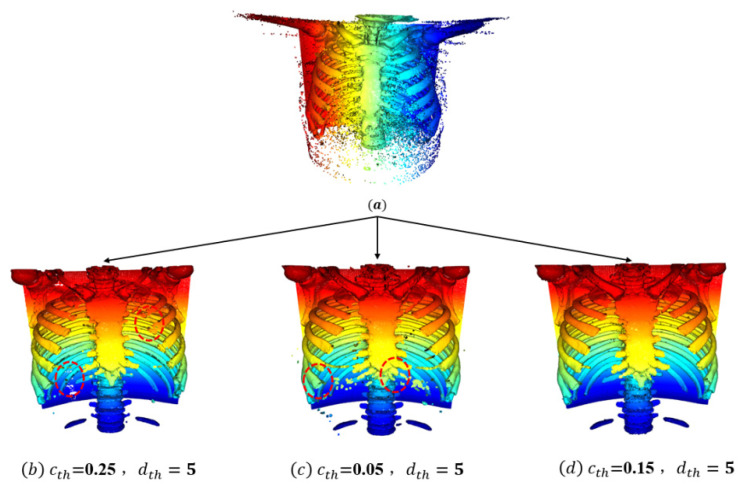
Effects of the Point Cloud Adaptive Smoothing Denoising Algorithm.

**Figure 7 sensors-24-04076-f007:**
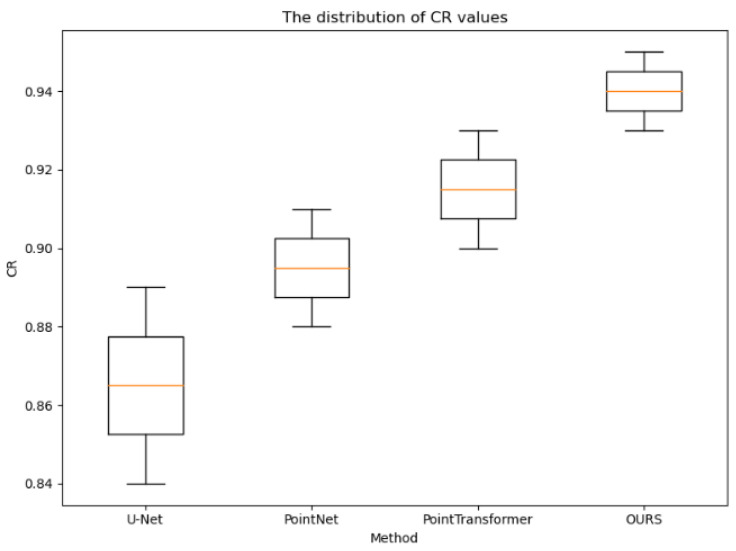
Distribution of CR values for all test samples of the four experimental methods.

**Figure 8 sensors-24-04076-f008:**
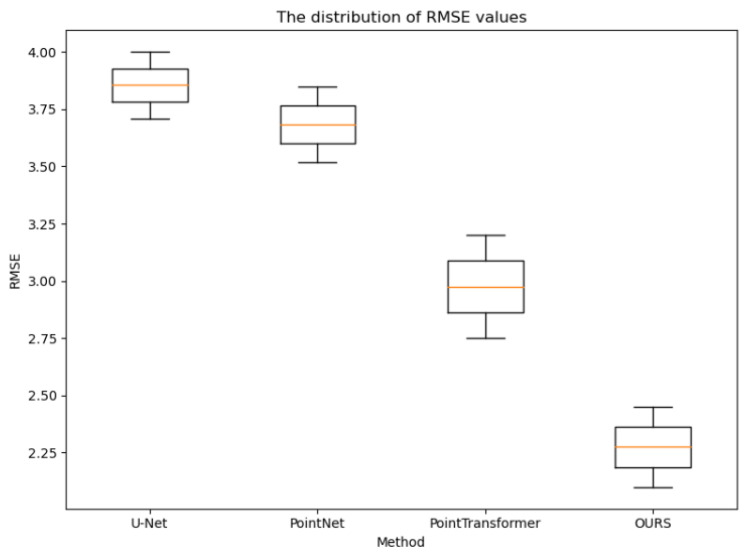
Distribution of RMSE values for all test samples of the four experimental methods.

**Figure 9 sensors-24-04076-f009:**
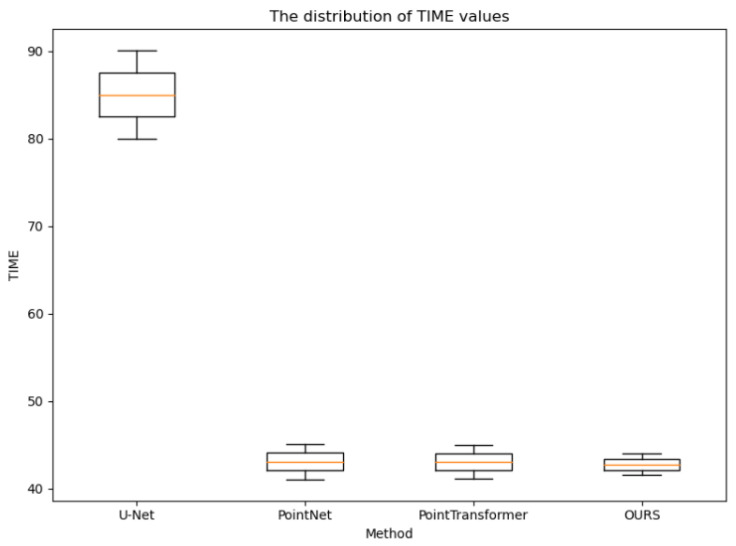
Distribution of TIME values for all test samples of the four experimental methods.

**Figure 10 sensors-24-04076-f010:**
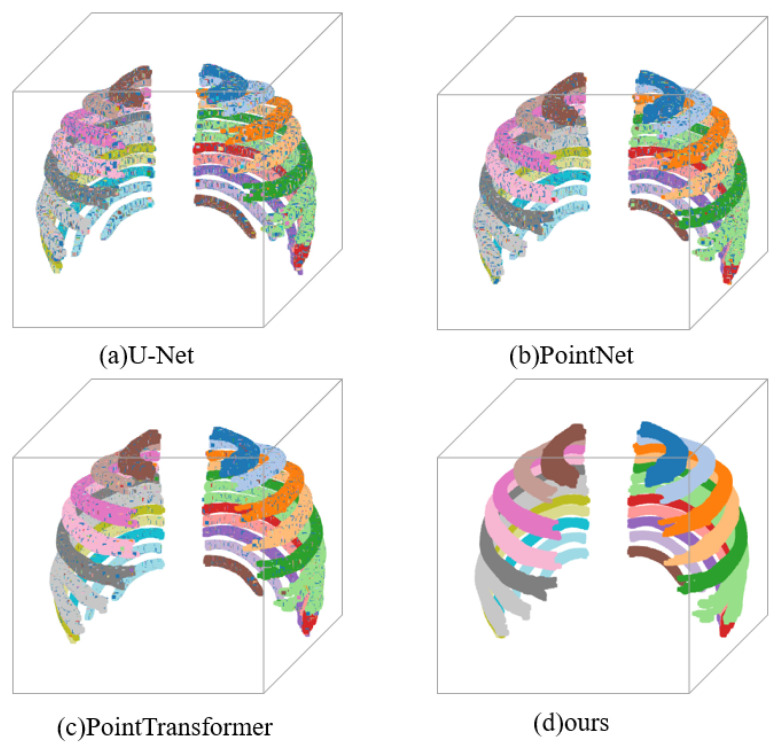
Three-dimensional reconstruction results of the ribs using four different experimental methods.

**Figure 11 sensors-24-04076-f011:**
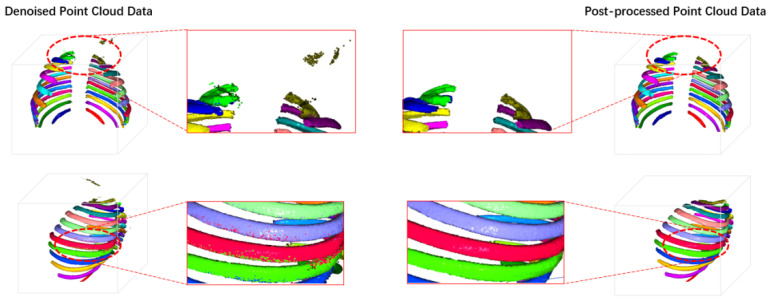
Post-Processing Effect of 3D Point Cloud Reconstruction.

**Table 1 sensors-24-04076-t001:** Dataset Partition Table.

Scheme	Number of CT Scans (Count)	Number of Point Clouds (Count)
Training set	290	725,080,000
Validation set	37	88,840,000
Testing set	37	0

**Table 2 sensors-24-04076-t002:** Metrics Table for the Point Cloud Adaptive Smoothing Denoising Algorithm.

Method Parameters	Mean CR	Mean RMSE
$c_{th}=0.25,d_{th}=4$	0.86	2.85
$d_{th}=5$	0.88	2.81
$d_{th}=6$	0.85	2.88
$c_{th}=0.15,d_{th}=4$	0.92	2.66
$d_{th}=5\;\;\;\;\;\;\;$	0.94	2.61
$d_{th}=6\;\;\;\;\;\;\;$	0.93	2.65
$c_{th}=0.05,d_{th}=4\;\;\;\;\;\;\;$	0.89	2.83
$d_{th}=5\;\;\;\;\;\;\;$	0.90	2.79
$d_{th}=6\;\;\;\;\;\;\;$	0.87	2.84

**Table 3 sensors-24-04076-t003:** Comparison of the Performance of the Four Methods.

Method	Mean CR	Mean RMSE (mm)	Mean TIME (s)
Voxel-based (U-Net)	0.87	3.86	85.01
Point Cloud-based (PointNet)	0.90	3.72	42.36
Point Cloud-based (PointTransformer)	0.91	2.97	42.45
Point Cloud-based (OURS)	0.94	2.28	42.58

**Table 4 sensors-24-04076-t004:** Performance Comparison Table of Post-Processing Stages.

Data Removal	Data Smoothing	Label Smoothing	Mean CR	Mean RMSE
			0.91	2.61
✓			0.92	2.49
✓	✓		0.93	2.39
✓	✓	✓	0.94	2.28

## Data Availability

The datasets analyzed during this study are obtained from public datasets.

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
