# Peer review of "Research on Three-Dimensional Reconstruction of Ribs Based on Point Cloud Adaptive Smoothing Denoising"

_sensors, 2024, doi:10.3390/s24134076_

Round 1
Reviewer 1 Report
Comments and Suggestions for Authors
This paper proposes a novel rib 3D reconstruction method based on point cloud adaptive smoothing and denoising. The method involves converting voxel data from CT images into multi-attribute point cloud data, followed by the application of adaptive smoothing and denoising techniques to eliminate noise and non-rib points. This approach, combined with efficient 3D reconstruction and post-processing techniques, aims to achieve high accuracy and comprehensive rib 3D reconstruction. While the work is well-organized and the results are promising, further improvements are necessary to enhance the method's effectiveness:
1- In equation (6), the term "sigma" needs to be clearly defined in the manuscript.
2- In Section "B. Experimental Setup" page 8, paragraph 1, Line 3, what is AdamW?
3- In the same section, the authors indicate that the learning rate is based on a warm-up cosine annealing method. What is the impact of this method in this work, and why is the interval set to [0.00001, 1]?
4- This study lacks a comparison with other works in the literature that employ different 3D reconstruction methods. The authors should perform a comparison to validate the proposed approach.
5- Additionally, the authors should enhance the "References" section with more recent citations in the same field.
Author Response
Reviewer: In equation (6), the term "sigma" needs to be clearly defined in the manuscript.
The author’s answer: We have added a definition and elaboration for the sigma parameter, and have provided the specific value we are using for this parameter in our research.
Reviewer: In Section "B. Experimental Setup" page 8, paragraph 1, Line 3, what is AdamW?
The author’s answer: We have introduced the AdamW optimizer and explained its advantages.
Reviewer: In the same section, the authors indicate that the learning rate is based on a warm-up cosine annealing method. What is the impact of this method in this work, and why is the interval set to [0.00001, 1]?
The author’s answer: We have outlined the advantages of the Cosine Annealing with Warm Restarts method and explained the reasons for using this method. We have also described the rationale behind the selection of the parameters for this method.
Reviewer: This study lacks a comparison with other works in the literature that employ different 3D reconstruction methods. The authors should perform a comparison to validate the proposed approach.
The author’s answer: We have introduced the latest literature methods in the field of rib segmentation and reconstruction, and have compared our method with these literature methods. We have also elaborated in detail on the advantages of our method over the literature methods.
Reviewer: Additionally, the authors should enhance the "References" section with more recent citations in the same field.
The author’s answer: We have introduced the latest 3D reconstruction papers in the related field in the references, and have elaborated on and introduced them in the related work section.
Reviewer 2 Report
Comments and Suggestions for Authors
The introduction could benefit from a more detailed explanation of the specific clinical implications of improved rib 3D reconstruction, such as potential impacts on patient outcomes.
The literature review could be enriched by adding more references about applications of artificial neural network, such as ‘Optimizing strength of directly recycled aluminum chip-based parts through a hybrid RSM-GA-ANN approach in sustainable hot forging’
Include a table summarizing key points of the discussed methods in the Literature Review section could enhance clarity.
Methodology: a flowchart of the entire process could help readers visualize the methodology more effectively.
Experiments: It would be helpful to include more qualitative results, such as visual comparisons of reconstructed models from different methods.
It could provide a more detailed discussion on future research directions, specifically how the proposed method could be further improved or extended to other types of medical imaging.
Author Response
Reviewer: The introduction could benefit from a more detailed explanation of the specific clinical implications of improved rib 3D reconstruction, such as potential impacts on patient outcomes.
The author’s answer: We have highlighted how the method presented in this paper can have a significant impact on clinical medical practice, helping to improve surgical success rates and shorten recovery times, among other advantages.
Reviewer: The literature review could be enriched by adding more references about applications of artificial neural network, such as ‘Optimizing strength of directly recycled aluminum chip-based parts through a hybrid RSM-GA-ANN approach in sustainable hot forging’
The author’s answer: We have added more references related to the applications of neural networks, thereby enriching the literature review.
Reviewer: Include a table summarizing key points of the discussed methods in the Literature Review section could enhance clarity..
The author’s answer: We have summarized the rib 3D reconstruction methods discussed into two main categories: the traditional graphics processing approaches based on machine learning, and the deep learning-based voxel reconstruction methods. We have also elaborated on the current applications of point clouds in medical imaging, which helps to clarify the differences between the various approaches and sets the stage for our proposed method.
Reviewer: Methodology: a flowchart of the entire process could help readers visualize the methodology more effectively.
The author’s answer: We have included Figure 1 which illustrates the overall workflow of our method, with corresponding visualizations for each stage. This helps to facilitate the understanding of the role and effects of each step in the method.
Reviewer: Experiments: It would be helpful to include more qualitative results, such as visual comparisons of reconstructed models from different methods.
The author’s answer: We have added Figure 10 to visualize the 3D reconstructed rib models obtained using different methods. By analyzing the reconstruction results of the various models, we are able to clearly demonstrate the limitations of the existing approaches and the advantages of our proposed method.
Reviewer: It could provide a more detailed discussion on future research directions, specifically how the proposed method could be further improved or extended to other types of medical imaging.
The author’s answer: We have included a detailed discussion on future directions, exploring the potential of extending our proposed method to other types of medical imaging data, such as CT and MRI, and investigating the 3D reconstruction performance across different imaging modalities.
Reviewer 3 Report
Comments and Suggestions for Authors
This manuscript introduces a new method for 3D reconstruction of ribs based on point cloud adaptive smoothing and denoising to overcome the limitations in reconstruction accuracy and computational efficiency, the research is interesting and provides valuable results, but the current document need major revision.
General considerations:
(1) The review of related research work in the introduction part could be more comprehensive and systematic to highlight the innovation of this study.
(2) The selection of parameters of the adaptive smoothing denoising algorithm lacks sufficient theoretical explanations and experimental comparative analyses.
(3) The comparative analyses with other latest methods in terms of reconstruction accuracy and efficiency are insufficient and need to be further supplemented.
(4) The selection of evaluation indexes is relatively single. At present, only two indicators, reconstruction accuracy and reconstruction time, are used, and it is possible to consider adding other evaluation indicators to make the evaluation more comprehensive.
(5) The experimental part lacks the discussion and analysis of model parameters, such as the basis for the selection of smoothness, denoising threshold and other parameters, and their influence on the results.
(6) For the completeness of the manuscript, the authors may add more articles on medical image analysis such as 3D reconstruction using point cloud data.
(7) For the references, the authors may add more state-of-art computer vision&3D point cloud articles in precision agriculture for the integrity of the manuscript (3D vision technologies for a self-developed structural external crack damage recognition robot; Automation in Construction.).
Author Response
Reviewer: The review of related research work in the introduction part could be more comprehensive and systematic to highlight the innovation of this study.
The author’s answer: In the introduction, we have provided a systematic review of the previous 3D rib reconstruction methods, which can be mainly categorized into traditional image processing methods (machine learning methods) and deep learning methods (voxel-based reconstruction methods). Our innovation primarily lies in introducing point cloud techniques into the field of 3D rib reconstruction, and leveraging point clouds to achieve high-quality rib reconstruction.
Reviewer: The selection of parameters of the adaptive smoothing denoising algorithm lacks sufficient theoretical explanations and experimental comparative analyses.
The author’s answer: We have added an elaboration on the parameter selection for the adaptive smoothing and denoising algorithm used in the rib reconstruction. We have also plotted 3D surface plots to clearly observe the influence of the corresponding parameters on the experimental results, which helps to quickly locate the appropriate parameter range.
Reviewer: The comparative analyses with other latest methods in terms of reconstruction accuracy and efficiency are insufficient and need to be further supplemented.
The author’s answer: We have added a description of the latest methods in the field of 3D rib reconstruction from recent years, and have compared our method with these latest approaches. Through the analysis of corresponding metrics and visualization results, we have elaborated on the advantages of our method.
Reviewer: The selection of evaluation indexes is relatively single. At present, only two indicators, reconstruction accuracy and reconstruction time, are used, and it is possible to consider adding other evaluation indicators to make the evaluation more comprehensive.
The author’s answer: We have introduced the Completeness Ratio metric, which can directly reflect the quality of the reconstruction, thus enhancing the comprehensiveness of the evaluation.
Reviewer: The experimental part lacks the discussion and analysis of model parameters, such as the basis for the selection of smoothness, denoising threshold and other parameters, and their influence on the results.
The author’s answer: Through the data in Figure 5 and Table 2, we have, while keeping the other stages of the method unchanged, compared the Mean CR and Mean RMSE metrics obtained under different parameter settings. This clearly demonstrates the influence of different parameters on the experimental results.
Reviewer: For the completeness of the manuscript, the authors may add more articles on medical image analysis such as 3D reconstruction using point cloud data.
The author’s answer: We have also added discussions on the literature regarding the use of point cloud data for 3D reconstruction in the medical field, ensuring the completeness of the manuscript.
Reviewer: For the references, the authors may add more state-of-art computer vision&3D point cloud articles in precision agriculture for the integrity of the manuscript (3D vision technologies for a self-developed structural external crack damage recognition robot; Automation in Construction.).
The author’s answer: In the references section, we have added more of the latest articles on computer vision and 3D point cloud research, ensuring the completeness of the manuscript.
Round 2
Reviewer 1 Report
Comments and Suggestions for Authors
The authors have followed all necessary recommendations.
Reviewer 3 Report
Comments and Suggestions for Authors
accept